# Resource sharing is sufficient for the emergence of division of labour

Jan J. Kreider [1] ✉, Thijs Janzen [1], Abel Bernadou [2,3], Daniel Elsner [1,4], Boris H. Kramer [1] & Franz J. Weissing [1] ✉

Division of labour occurs in a broad range of organisms. Yet, how division of labour can emerge in the absence of pre-existing interindividual differences is poorly understood. Using a simple but realistic model, we show that in a group of initially identical individuals, division of labour emerges spontaneously if returning foragers share part of their resources with other group members. In the absence of resource sharing, individuals follow an activity schedule of alternating between foraging and other tasks. If non-foraging individuals are fed by other individuals, their alternating activity schedule becomes interrupted, leading to task specialisation and the emergence of division of labour. Furthermore, nutritional differences between individuals reinforce division of labour. Such differences can be caused by increased metabolic rates during foraging or by dominance interactions during resource sharing. Our model proposes a plausible mechanism for the self-organised emergence of division of labour in animal groups of initially identical individuals. This mechanism could also play a role for the emergence of division of labour during the major evolutionary transitions to eusociality and multicellularity.

Division of labour—within-individual consistency but between-individual variability in task choice among the members of a group[1]— is a pivotal aspect of social life in human and animal societies. Division of labour occurs in a broad range of organisms[2,3]. For instance, in eusocial insects, workers specialise in foraging, defending the nest, or nursing the brood[4]. In some species of birds, such as noisy miners, some helpers specialise in chick provisioning or mobbing nest predators[5]. In the Lake Tanganyika princess cichlid, helpers engage in predator defence, nest maintenance or egg care[6,7]. Within prides of lionesses, individuals perform different roles during hunting and territorial defence, and they can specialise to take care of the cubs[8].

A central question for understanding social life is consequently how such division of labour can originate. The emergence of division of labour is typically modelled in response threshold models[1,9–14]. These models assume that individuals differ in thresholds that determine their likelihood to start performing a task when they perceive a task stimulus. Individuals with a low response threshold take on a task more readily, thereby decreasing the task stimulus; this prevents other individuals with a higher threshold from taking on the task as well. As a result, individuals specialise in those tasks for which they have a low threshold. However, task specialisation and division of labour only occur if there are pre-existing interindividual differences in response thresholds[15]. It remains poorly understood how self-organised division of labour can emerge within homogenous groups of highly similar or even identical individuals.

Here, we present a model for the emergence of division of labour in animal groups of identical individuals. The model makes the realistic assumption that the nutrition level of individuals declines over time, and that a low nutrition level triggers foraging[16–22] (Fig. 1a). In the absence of interindividual interactions, one would expect that individuals sequentially perform non-foraging behaviours (henceforth referred to as nursing) when

[1]Groningen Institute for Evolutionary Life Sciences, University of Groningen, Nijenborgh 7, 9747 AG Groningen, The Netherlands. [2]Zoology / Evolutionary Biology, University of Regensburg, Universitätsstraße 31, 93053 Regensburg, Germany. [3]Centre de Recherches sur la Cognition Animale (CRCA), Centre de Biologie Intégrative (CBI), Université de Toulouse, CNRS, UPS, 31062 Toulouse, France. [4]Department of Evolutionary Biology and Ecology, Institute of Biology I (Zoology), University of Freiburg, Hauptstraße 1, 79104 Freiburg (Breisgau), Germany. ✉e-mail: j.j.kreider@rug.nl; f.j.weissing@rug.nl

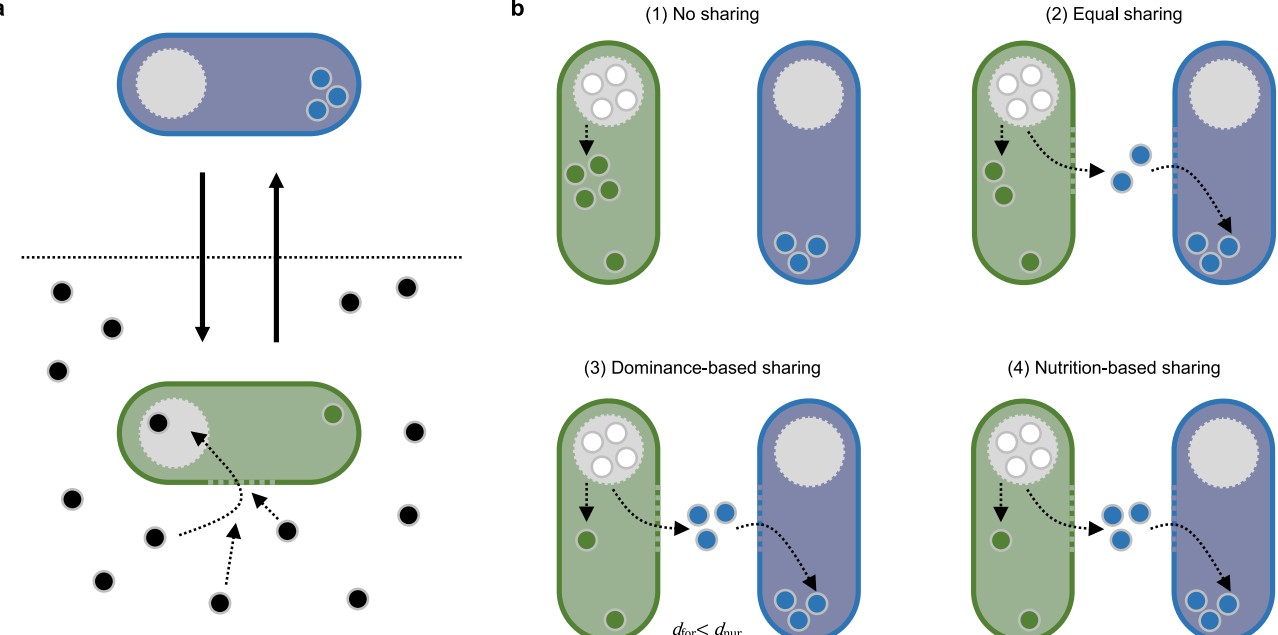

**Fig. 1 | Task switching and four resource sharing scenarios. a** Our model assumes that individuals can switch (solid arrows) between two states: foraging (green) and nursing (blue). While foraging, individuals retrieve resources (black dots) from the environment and store these resources in a temporary storage organ (grey circle). As long as resources are in the storage organ, they can in principle be shared with other individuals. Once they are integrated into the individual's body (green dots in foraging individuals; blue dots in nursing individuals) they contribute to the individual's nutrition level and can no longer be shared. As foraging is triggered by a low nutrition level, foraging individuals have on average lower nutrition levels than nursing individuals. **b** Four resource sharing scenarios considered in our study: (1) No sharing: Resources are not shared; foraging individuals consume all resources themselves. (2) Equal sharing: Foraging individuals share the collected resources equally between themselves and a nursing individual. (3) Dominance-based sharing: Each individual has a pre-assigned dominance value and resources are shared in relation to these dominance values: The dominant individual obtains a larger proportion of the collected resources. (4) Nutrition-based sharing: As in (3), but now the dominance level of an individual is not pre-assigned and constant, but proportional to the individual's nutrition level.

nutrition levels are high and foraging behaviour when nutrition levels are low. However, within groups of animals, individuals interact and resource sharing regularly occurs[23,24]. This could lead to a certain degree of specialisation on foraging and non-foraging behaviour, as foragers that give part of their resources away may soon have to forage again, while the receivers of the shared resources can delay foraging. We therefore considered various resource-sharing scenarios (Fig. 1b), in order to investigate whether this common mechanism is sufficient to generate a high degree of task specialisation and division of labour.

## Results

### Resource sharing is sufficient for the emergence of division of labour

We ran 20 replicate simulations for each resource sharing scenario. We quantified the degree of division of labour that emerges in the simulations using the metric $D$ that was introduced by Duarte et al.[25] (see Methods for details and Fig. S1 in the Supplement for other division-of-labour metrics). For the scenarios considered here (see Methods), $D$ ranges from −1 to +1, where −1 indicates the strict alternation between tasks, 0 indicates random switching between tasks, and +1 indicates task specialisation[25]. Figure 2a shows that in the absence of resource sharing (Fig. 1b; (1) No sharing), the division-of-labour metric $D$ takes on the extreme value −1, implying that individuals follow an activity schedule of alternating between tasks. Nursing individuals that start to forage to obtain resources keep all the resources for themselves. Consequently, they nurse until their nutrition level has dropped to a threshold that induces them to start foraging again. As shown in Fig. 2b, the outcome is very different when resources are shared between foraging and nursing individuals (Fig. 1b; (2) Equal sharing). Now the metric $D$ reaches positive values above 0.5,

indicating that individuals specialise on foraging or nursing for longer stretches of time.

### Division of labour is reinforced if foragers have a higher metabolic rate

As metabolic rates can differ with task[26,27], we tested how such differences affect the emergence of division of labour. Even if the metabolic rate associated with nursing is only slightly lower (90% or 95%) than the metabolic rate associated with foraging, division of labour is strongly reinforced (Fig. 3a), leading to a bimodal distribution of nutrition levels (Fig. 3b). If metabolic rates associated with nursing are higher compared to foraging, division of labour does still occur, but it becomes weaker. Similarly, when the duration of nursing is shorter than the duration of foraging, division of labour is reinforced because this decreases the amount of energy that nursing individuals metabolise during task performance relative to foraging individuals (Fig. S2).

### Dominance relationships reinforce division of labour, especially when dominance is related to nutritional status

Resource sharing is often not egalitarian but associated with dominance[18,28–30]. In order to investigate uneven resource sharing through dominance effects, we assigned constant dominance values to the individuals in the simulation at initialisation (Fig. 1b; (3) Dominance-based sharing). Under such dominance-based differences in resource intake, division of labour reaches maximal levels and nutrition levels of foraging and nursing individuals are bimodally distributed (Fig. 4a1). As individuals with larger dominance values obtain more resources during sharing, they rapidly are fixed into nursing whereas those with low dominance values consistently forage. Individuals with intermediate dominance values either forage or nurse (Fig. 4a2), which is stochastically driven by the random interactions

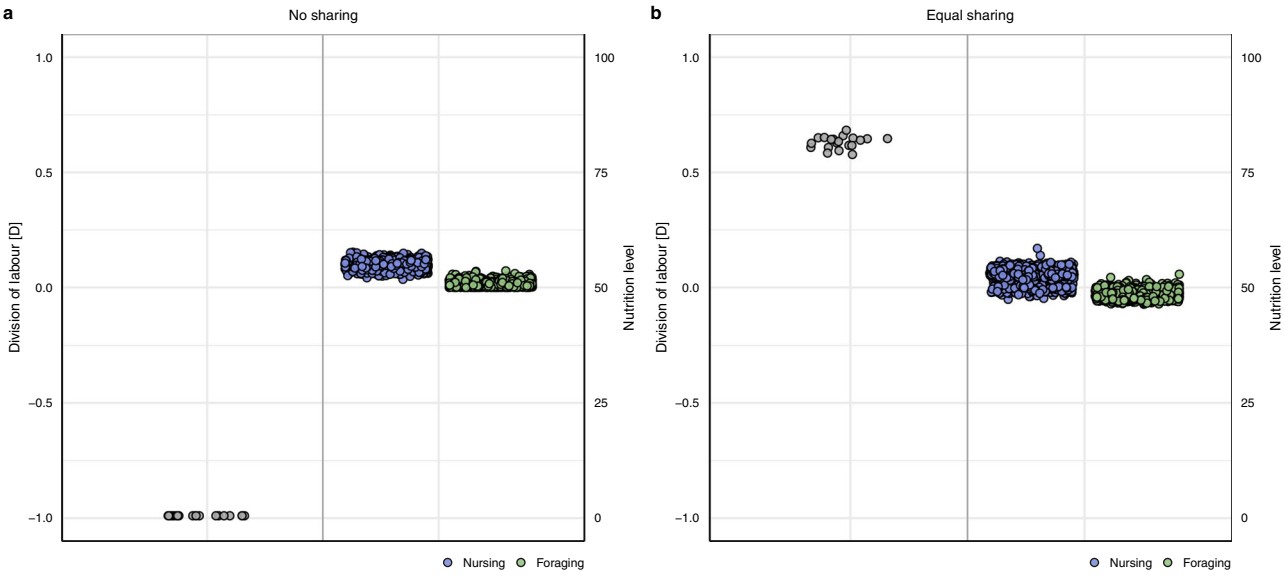

**Fig. 2 | Division of labour and nutrition levels of nursing and foraging individuals under no resource sharing and equal sharing. a** No sharing. In the absence of resource sharing, $D = -1$ which indicates that individuals alternate between nursing and foraging. As foraging is triggered by a decline in the nutrition level, nursing individuals have a higher nutrition level than foragers. **b** Equal sharing. If foraging individuals share their collected resources equally with nursing individuals, $D \approx 0.6$ which represents an intermediate level of division of labour. Each grey dot (left panel) is the division of labour metric from a replicate simulation ($n = 20$). Blue (nursing) and green (foraging) dots (right panel) represent the nutrition levels of all individuals at the end of all replicate simulations.

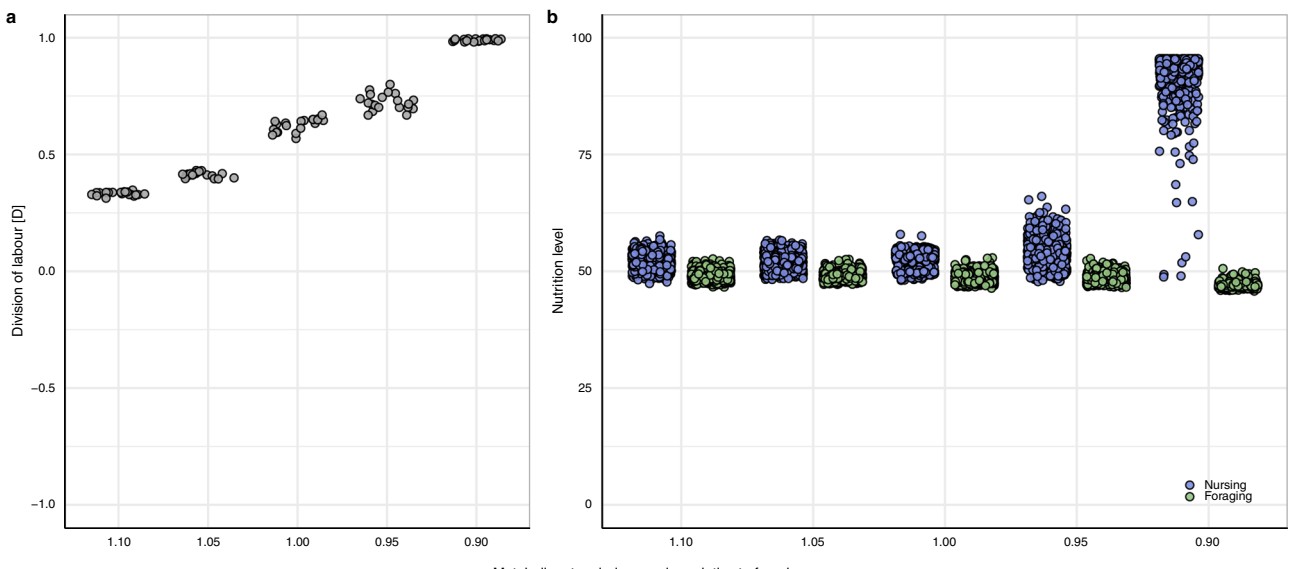

**Fig. 3 | Division of labour when the metabolic costs of foraging and nursing differ. a** If metabolic rates during nursing are lower relative to foraging, division of labour is reinforced. Conversely, division of labour is weaker when metabolic rates during nursing are higher than during foraging. **b** As their metabolic rates become lower, the nutrition level of nursing individuals deviates more strongly from those of foraging individuals. For graphical conventions, see Fig. 2.

that those individuals have at the start of a simulation. The model scenario considered in Fig. 4a has the drawback that division of labour is based on pre-existing differences (in dominance) between individuals. However, the same result can also be obtained in a population of initially identical individuals, if dominance is caused by differences in nutrition level (Fig. 1b; (4) Nutrition-based sharing). As shown in Fig. 4b1, division of labour again reaches maximal levels, and the nutrition levels of foraging and nursing individuals are bimodally distributed. At the start of the simulation, nutrition levels of individuals diverge relatively slowly but once small differences are present, a full divergence of nutrition levels is rapidly achieved (Fig. 4b2). In Fig. S3, we show that division of labour even emerges if individuals with low

nutrition levels obtain more resources during sharing. In this more altruistic sharing scenario, low levels of division of labour emerge because some individuals that are about to start foraging obtain resources from returning foraging individuals and thus delay foraging. This result is similar to that in Fig. 3, where low levels of division of labour even emerge if foraging is associated with lower metabolic rates than nursing.

**Division of labour emerges irrespective of group size**
Task specialisation has been suggested to increase with group size[12,31,32] but this is not supported unequivocally[25,33]. We therefore altered group size to investigate its effect on the emergence of division of labour.

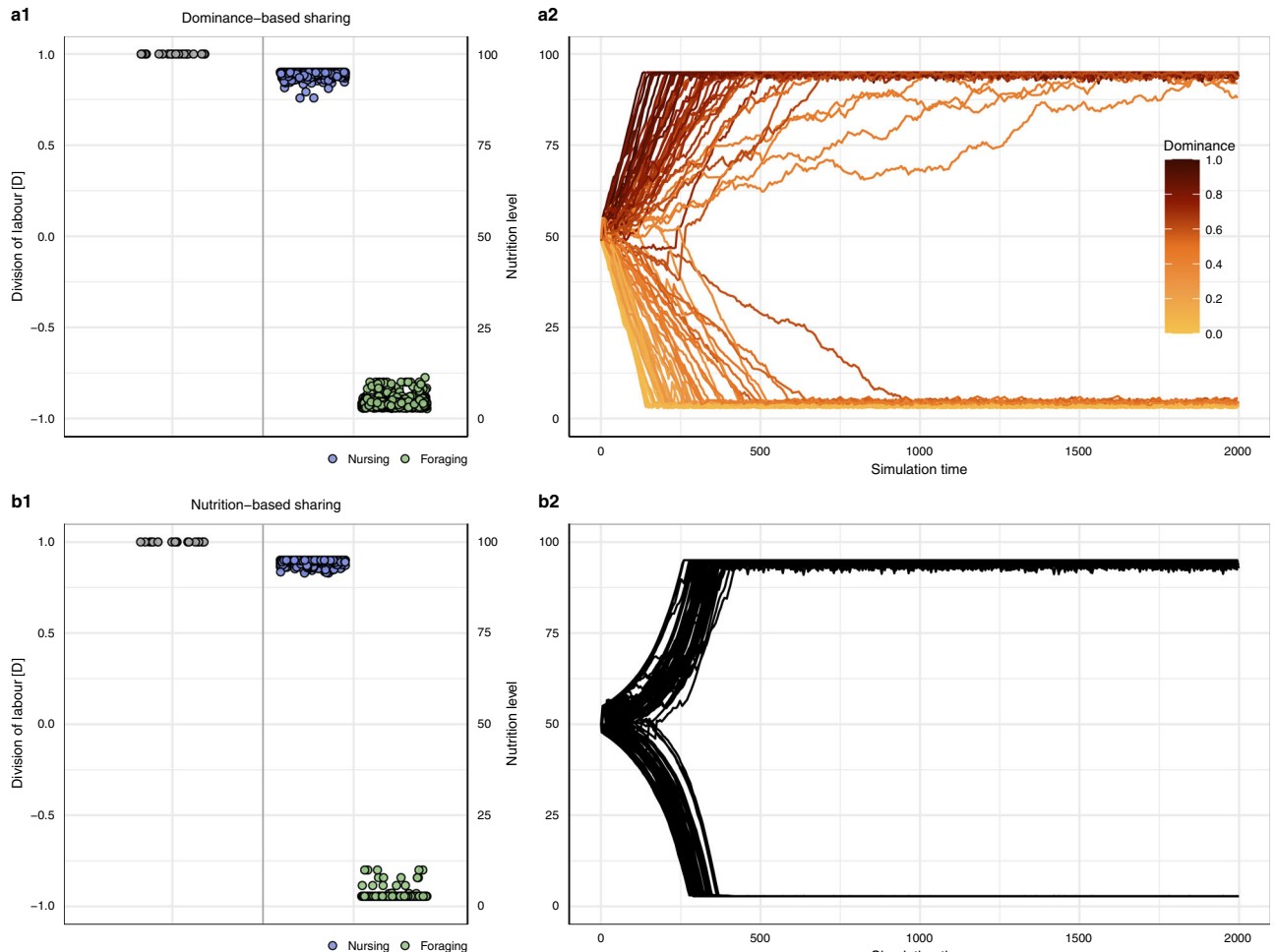

**Fig. 4 | Division of labour, nutrition level and dominance of nursing and foraging individuals. a1** Dominance-based sharing. Division of labour reaches maximal levels of $D = 1$. Nursing and foraging individuals exhibit a bimodal distribution of nutrition levels. **a2** Nutrition levels of individuals diverge over simulation time, depending on the individual's dominance. **b1** Nutrition-based sharing. Division of labour again reaches maximal levels of $D = 1$, and nutrition levels exhibit a bimodal distribution. **b2** Nutrition levels of individuals diverge over simulation time. **a1 + b1** For graphical conventions, see Fig. 2. **a2 + b2** Each line shows the nutrition level of an individual over the first 20% of simulation time from a representative replicate simulation, coloured according to the individual's dominance value if dominance values were pre-assigned.

Figure 5 shows that division of labour emerges independently of group size. However, in small groups, the degree of division of labour may strongly depend on details (such as whether group size is even or odd). Figure S4 further elaborates on this.

## Discussion

We have demonstrated that resource sharing can result in strong task specialisation and division of labour. Pre-existing differences between individuals can enhance division of labour, but they are not required for its emergence. Our model is based on two general and plausible assumptions. First, individual nutrition levels decline in the course of time, inducing individuals to go out foraging once they become very low. In the absence of interactions with other individuals, this results in an activity schedule where individuals alternate between foraging (when nutrition levels are declining) and nursing (when nutrition levels are replenished). If, second, foragers share some of their collected resources with other individuals, this activity schedule becomes interrupted due to two feedbacks: nursing individuals may be fed before perceiving a hunger signal, thus retarding their next foraging bout; and foraging individuals cannot use the shared resources to fully replenish their nutrition level, thus reducing the time until the hunger signal induces them to start foraging again. These feedbacks lead to task specialisation and division of labour, because nursing individuals

that are fed by others are likely to continue nursing whereas foraging individuals, who feed others, are likely to proceed with foraging.

Most response threshold models assume that individuals differ in their response thresholds throughout their lifetime. These differences are typically assumed to be established early in life, either arising from genetic differences[9–13] or from stochastic processes during individual development[34–36]. In contrast, the individuals in our model all have the same threshold throughout their lifetime—there is stochasticity in our model as well, but it only affects the momentary decision of individuals and not the state of their decision mechanism. This difference between our model and the more traditional models could be tested experimentally. If division of labour emerges due to stable interindividual differences in response thresholds, as assumed by response threshold models, an individual that is foraging (or nursing) in one social group would tend to forage (or nurse) again when placed into another group. If, in contrast, all individuals employ the same decision mechanism and division of labour arises through feedback mechanisms acting on other parameters (such as individual nutrition level, as in our model), no such correlation is to be expected if the nutritional level (or any other feedback parameter) is equalised at the start of each experiment.

In some models of microbial specialisation[34,35], the first step of specialisation results from phenotypic noise, that is, from stochastic factors that break the symmetry among initially identical individuals by

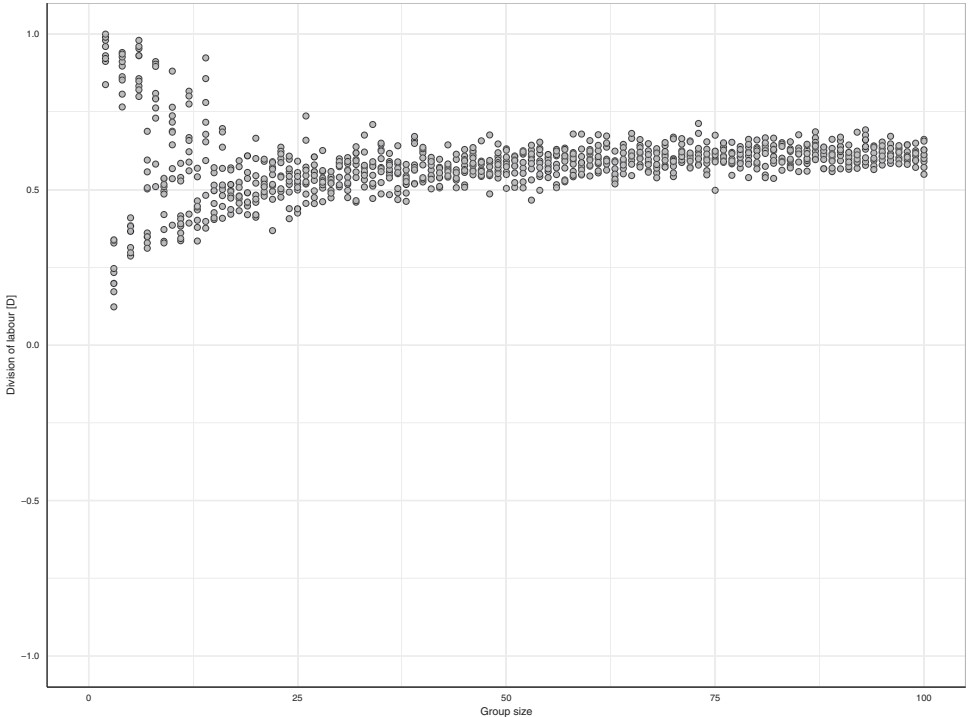

**Fig. 5 | Effect of group size on division of labour in the equal sharing scenario.** Division of labour emerges irrespective of group size. Each dot shows the division-of-labour metric from a replicate simulation per group size ($n = 10$).

inducing small, random differences. Subsequently, these differences are stabilised and enhanced by specific properties of the system or by positive feedback processes. Our model also includes phenotypic noise as a symmetry breaker (as individuals assess their nutrition level with some inaccuracy), but as shown in Fig. 2a this in itself does not induce division of labour. For this, resource sharing is required (Fig. 2b). Resource sharing also plays a key role in models for microbial specialisation[34,37]. However, in these models, resources are globally available and thereby shared by individuals, whereas in our model individuals assess their nutrition level—an individual-level property—in order to decide their task, and resources are shared between specific individuals.

Several studies have proposed that division of labour in social insects is associated with nutritional differences between foragers and nurses[16–22,38–40]. In our model, division of labour leads to a situation where nutrition levels of nurses are higher than those of foragers, and division of labour is enhanced when the metabolic rate while foraging is higher than the metabolic rate while nursing. However, pre-existing nutritional differences are not required for the emergence of division of labour—the crucial ingredient is resource sharing. In social insects, resources can, for instance, be transferred mouth-to-mouth via trophallactic exchanges[24]. However, for the emergence of division of labour it is not required that resources are shared altruistically. Foragers could, for instance, deposit part of the resources in the nest (destined to be fed to the brood at a later stage), where they are consumed by other individuals instead.

We here employed a definition of division of labour based on within-individual consistency and between-individual variability in task choice among the members of a group[1]. This definition allows for the possibility that division of labour emerges spontaneously from mechanisms that have not been evolutionarily selected for regulating division of labour but for other purposes. Other definitions of division of labour additionally highlight the relevance of selection for cooperation between individuals[3]. In models for the evolution of such cooperative division of labour, the shareability of goods/benefits is essential for the evolution of division of labour[37,41]. Our model does not include

evolution, as the mechanisms underlying the self-organised emergence of division of labour (hunger-induced foraging and resource sharing) are assumed to be pre-existent and do not change during our simulations. Yet, such non-evolved division of labour can have important evolutionary implications, as it can be the substrate of subsequent adaptive evolution. This is elegantly shown in an experiment on cooperating robots[42], where adaptive division of labour evolved more easily in the presence of pre-existing behavioural differences. Similarly, some evolutionary theories for the evolution of division of labour in social insects suggest that the initial steps are based on pre-existing mechanisms that regulated behaviours in solitary ancestors[43–45].

Our model furthermore highlights the importance of dominance interactions and task-specific differences in nutrition level for the emergence of strong non-reproductive division of labour. Dominance and nutrition level also play an important role in the determination of reproductive abilities in social insects[28–30,46,47] and in social spiders[48], and thus in reproductive division of labour. For instance, in paper wasps, breeder and helper roles are likely determined by nutrition[49], as also indicated by the differential expression of storage proteins in breeders and helpers[50,51]. Furthermore, in eusocial insects, caste is often determined by a nutrition-dependent developmental switch[52], and across eusocial insects, castes differentially express genes coding for the storage protein vitellogenin[40,53–56]. Thus, differences in nutrition level that emerge in our model due to dominance interactions during resource sharing could also play an important role for the evolution of eusociality.

Response threshold models and empirical studies[12,31,32] (though not all[25,33]) suggest that task specialisation and division of labour are most pronounced in large groups. In our model, division of labour emerges even in very small groups. In fact, division of labour can be stronger in very small groups (e.g. groups of two) than in larger groups. Equivalently, experiments have shown that division of labour emerges in paired associations of ants[18,57], and the sexes of some species of birds exhibit strong division of labour between the breeding female and the foraging male[58–60]. Consequently, the strength of division of labour does not necessarily depend on group size alone but also the details of

social interactions, such as the number of individuals with which foragers share resources, are crucial to be considered.

Lastly, resource sharing could be a mechanism for the emergence of division of labour beyond animal groups. For instance, in some cyanobacteria, cells are specialised on obtaining different resources—carbon or nitrogen—for the colony[61]. Resource sharing could thus play a role in the emergence and regulation of cellular division of labour during the major evolutionary transition to multicellularity[62,63].

Overall, our model demonstrates that self-organised division of labour emerges through resource sharing between identical individuals. Given the omnipresence of resource sharing in biological systems—for instance, in group-living animals, eusocial insects, human hunter-gatherer societies, or within multicellular organisms—our model suggests a mechanism that could explain the emergence of division of labour across a broad range of organisms.

## Methods

### The model
We developed an individual-based simulation model in continuous time. Each simulation represents a group of $N$ individuals ($N = 100$, unless stated otherwise). Individuals either forage to obtain resources or perform other tasks to which we refer as nursing. Each simulation had a duration of $T$ time steps ($T = 10,000$).

### Nutrition level of individuals
Individuals possess an internal state variable $n$ that reflects their nutrition level and ranges from 0 to $n_{max}$ ($n_{max} = 100$). At the start of a simulation, all individuals are initialised with an identical nutrition level of $n_{init}$ ($n_{init} = 50$). Thus, simulations start with identical individuals. Over time, the nutrition level of individuals decreases by a metabolic rate $m_{for}$ when foraging, and $m_{nur}$ when nursing ($m_{for} = 1.0$, $m_{nur} = 1.0$, unless stated otherwise).

### Task choice
An individual's choice to forage or nurse depends on the individual's nutrition level. We assume that individuals become more likely to forage as their nutrition level declines. On average, individuals start foraging when their nutrition level reaches the critical threshold value $\mu$ ($\mu = 50$). However, we assume that individuals are not perfectly accurate in assessing their nutrition level: the perceived nutrition level is drawn from a normal distribution around the true nutrition level, with standard deviation $\sigma$ ($\sigma = 1$). After returning from a foraging trip, which has a fixed duration of $t_{for}$ time steps ($t_{for} = 5$), an individual has the choice between foraging again or to switch to nursing. The choice made depends on the individual's perceived nutrition level in relation to the threshold value for foraging: the individual will forage again if the perceived nutrition level is below the threshold, and it will switch to nursing otherwise. Resources can be received through resource sharing by nursing individuals, which include previously foraging individuals that refilled their nutrition levels and switched back to nursing. Individuals thus have no pre-disposition to perform nursing or foraging tasks and all individuals can in principle share and receive resources. If a nursing individual is not fed before its perceived nutrition level declines below its threshold value, then it starts to forage. If the nursing individual is fed by a foraging individual, then it processes the food for a fixed duration of $t_{nur}$ time steps ($t_{nur} = 5$), during which it cannot receive any further resources. After food processing, nursing individuals assess their nutrition level and decide whether to continue nursing or start foraging (Fig. S5).

### Nutrition transfers
Foragers obtain $R$ resources while foraging ($R = 10$). We implemented four different model scenarios of resource sharing. (1) No sharing: Upon return from a foraging trip, foraging individuals keep all of the resources they obtained. (2) Equal sharing: Upon return from a foraging trip, foraging individuals evenly share resources with $i$ nursing individuals. The number of resources that the individuals engaged in the interaction obtain is thus $R/(1 + i)$ because one foraging individuals interacts with $i$ nursing individuals. We assume that $i = 1$ but relax this assumption in the supplementary materials (Fig. S4). (3) Dominance-based sharing: Individuals are assigned a dominance value upon initialisation of the simulation, randomly sampled from a uniform distribution ranging from 0 to 1. The number of resources obtained by the individuals during the interaction is dependent on their dominance value, such that more dominant individuals obtain more resources than less dominant individuals. The number of resources obtained by individual $k$ is calculated using the softmax function

$$S_k = \frac{\exp(d_k s)}{\exp(d_k s) + \sum_{l=0}^{i} \exp(d_l s)} \quad (1)$$

where $d_k$ is the dominance value of focal individual $k$, $d_l$ is the dominance value of the other $i$ individuals and $s$ is a parameter for the softmax function. If $s = 0$, the model reduces to the equal sharing scenario, if $s < 0$, individuals with the lowest dominance values receive a disproportionate number of resources, and if $s > 0$, individuals with the highest dominance values receive a disproportionate number of resources. For the simulations with a dominance effect, we used $s = 1$, which results in the most dominant individual getting slightly more than their expected share based on their relative dominance. (4) Nutrition-based sharing: The dominance status of an individual is given by its nutrition level, relative to the maximum nutrition level possible, i.e. $d_k$ is now given by $n/n_{max}$. Otherwise, interactions proceed as in scenario 3, using Eq. (1) to determine resource distribution among individuals. Again, we used $s = 1$ (for simulations with other values of $s$, see Fig. S3). As dominance again determines resource distribution among individuals, individuals who have a higher nutrition level obtain more resources than individuals with a lower nutrition level.

When a foraging individual returns from a foraging trip, but no nursing individuals who can obtain resources are available, the foraging individual consumes all of the resources itself, independently of the sharing scenario.

### Quantifying division of labour
We here defined division of labour as within-individual consistency but between-individual variability in task choice among the members of a group[1]. As within-individual consistency and between-individual variability are difficult to quantify in a single metric[64], we used four different metrics for quantifying the degree of division of labour that emerges in the simulations[25,31,64]. The four metrics place different emphasis on the two defining properties of division of labour, but as shown in Fig. S1 they yield similar results for the main model scenarios. In the main manuscript, we therefore only report the results of the metric $D$ proposed by Duarte et al.[25], which is easy to understand and calculate. To calculate $D$, one considers all task choice situations (see Fig. S5, for when task choice situations occur), where individuals have to decide whether to perform the previous task again or to switch to the other task. It is defined as:

$$D = \frac{\bar{q}}{p_1^2 + p_2^2} - 1. \quad (2)$$

where $\bar{q}$ is the proportion of cases that individuals chose to perform the previous task again (averaged over all individuals and choice situations), while $p_1$ and $p_2 = 1 - p$ are the relative frequencies with which each of the two tasks (foraging and nursing) are chosen (averaged over all individuals and choice situations). The term

$p_1^2 + p_2^2$ in the denominator of Eq. (2) corresponds to the probability to randomly choose to perform the previous task again. Hence, $D$ is positive if $\bar{q} > p_1^2 + p_2^2$, that is, if individuals have a higher than random tendency to perform the previous task again. As $\bar{q}$ ranges between 0 and 1, $D$ ranges between −1 (when $\bar{q} = 0$) and $2p_1 p_2 / (p_1^2 + p_2^2)$ (when $\bar{q} = 1$). In our model, the two tasks were approximately performed with equal frequency in all scenarios ($p_1 \approx p_2$). As a consequence, $D$ ranged between −1 (indicating alternation between tasks) and +1 (indicating full task specialisation)[25]. We calculated all division-of-labour metrics over the last 10% of simulation time to avoid measuring initialisation effects.

## Model analysis

The model was implemented in C++ and compiled with g++ 9.3.0. Model results were analysed and visualised in R 4.1.0[65] using the packages *ggplot2*[66], *gridExtra*[67], *cowplot*[68] and *MetBrewer*[69].

## Reporting summary

Further information on research design is available in the Nature Portfolio Reporting Summary linked to this article.

## Data availability

All data generated during this study is available under https://doi.org/10.34894/WD4EJZ.

## Code availability

Simulation code and data analysis scripts are available under https://doi.org/10.34894/WD4EJZ.

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

## Acknowledgements

We thank Hanno Hildenbrandt for helping us debug the model and increase simulation efficiency. We thank the Center for Information Technology of the University of Groningen for providing access to the Peregrine high performance computing cluster. J.J.K. was supported by an Adaptive Life grant by the University of Groningen. A.B. was supported by DFG grant no. BE6684/1–1 (Research Unit "So-long" - FOR 2218). D.E. was supported by DFG grant no. KO1895/19–1 (to Judith Korb) and by NWO grant no. 823.01.006 (to F.J.W.). F.J.W. acknowledges funding from the European Research Council (ERC Advanced Grant No. 789240).

## Author contributions

Conceptualisation: J.J.K., T.J., A.B., D.E., B.H.K., F.J.W.; Implementation: T.J., J.J.K.; Model analysis: J.J.K., D.E.; Writing: J.J.K., T.J., A.B., D.E., B.H.K., F.J.W.

## Competing interests

The authors declare no competing interests.
