## [Peer Review File · Nature Communications]

Resource sharing is sufficient for the emergence of division of labourReviewers' Comments:

Reviewer #1:

Remarks to the Author:

This paper uses simulations to lay out different scenarios in which division of labour can emerge spontaneously from groups with identical individuals. They find that resource sharing is sufficient for task specialisation to occur between "foragers" and "nurses". They find that higher levels of task specialisation occur if: 1. there is a higher metabolic cost to foraging, 2. if there is a dominance hierarchy between individuals, and 3. if more nutritionally starved individuals receive more resources when foragers share.

This is a well developed model with some interesting findings. However, the framing of the model and the context of its structure and results within the wide body of theory could be improved. I make below several key suggestions for the framing of this paper.

First, I found the paper's definition of division of labour (line 35) to be unsatisfactory. The field is currently trying to move to more precise definitions (for instance their references 2 and 13) and the definition proposed here is both unintuitive and vague. For instance, no mention is made that individuals must perform distinct tasks. Connecting the results of this paper to other theory first requires a clear understanding of how the definition of division of labour in this paper may compare or contrast to the definition of other works. For instance, I think that they are using a more descriptive, less evolutionary definition of division of labour more similar to reference 2 than reference 13, but this distinction is not made clear in the manuscript. I would like this to be more clear/explicit.

Secondly, this paper focuses on previous theory that uses response-threshold models and does not discuss the wider body of theory using alternative mechanisms, of which response threshold models are just a subset. For instance, there has been recent work discussing the broader variety of mechanisms by which division of labour can evolve (Cooper et al. "The evolution of mechanisms to produce phenotypic heterogeneity in microorganisms." *Nature Communications* (2022)). In particular, there is also work on how division of labour can emerge from "phenotypic noise" (Ackermann et al. "Self-destructive cooperation mediated by phenotypic noise." *Nature* 454.7207 (2008): 987-990.). Most notably, the paper "Evolution of self-organized task specialization in robot swarms." By Ferrante, et al. *PLoS computational biology* (2015) is not cited which seems to have many similarities in set-up and results to this work and so a thorough discussion of differences in assumptions and findings with this work is needed.

Third, I think that there are a couple key aspects of the model that are not discussed which would help to understand why the results emerge. First, there is an asymmetry between the roles built into the model where foragers share resources only with nurses and not other foragers. Do the results of the model work if foragers share benefits with all individuals regardless of their role? I think the connection to the above Ferrante paper would be clearer if this were expounded upon. Second, stochastic noise is built into the model by the internal threshold that individuals use to determine if they begin foraging. This seems to be a clear way that symmetry between individuals is broken in the model but is not sufficiently discussed. Does the model work without this stochastic component? In a sense this would make this mechanism a form of division of labour that is already expounded upon in the literature on phenotypic noise (the above Ackermann paper and Bistability in bacteria by Dubnau in *Molecular Microbiology* 2006). I think further discussion of this connection is warranted.

Finally, much of the theoretical literature in the area has asked why division of labour might evolve over uniform cooperation in social groups. This model suggests that division of labour should arise spontaneously from uniform cooperation due to resource/benefit sharing. Yet, we know empirically that this is not always the case. For instance, *Pseudomonas aeruginosa*, where any phenotypic heterogeneity in siderophore production arises from cheating not division of labour benefits (Jiricny et al. Fitness correlates with the extent of cheating in a bacterium. *J. Evol. Biol.* (2010)). Discussing why

shared resources may not lead to division of labour in some cases is needed.

Reviewer #2:

Remarks to the Author:

Thank you for the opportunity to review the manuscript "Resource sharing leads to the emergence of division of labour" by J.J.Kreider and colleagues.

In this work, the authors consider a population of agents switching tasks between nursing and foraging. Execution of both tasks consumes resources, which can be gained by performing foraging. Resources can be shared between individuals. Authors consider a number of well-designed scenarios of resource sharing: a negative control case of no sharing, the simplest equal sharing scenario, and two more elaborate cases - dominance-based and nutrition-based scenarios. Using this model, the authors perform simulations of the population dynamics and quantify the degree of the division of labour using the chosen metric. They show that the more elaborate scenarios of resource sharing results in more specialization. They also consider a number of variants of their model screening through the control parameters: group size and metabolic costs of the tasks.

The work addresses a fundamental question in biology - the emergence of specialization in a collective of initially identical organisms. The manuscript is well-written and very accessible. The model is simple enough to be easily comprehended but rich enough to address the problem in the focus of the study. Hence, there is a lot of merit in this work. However, do not find the presented results convincing: partially due to the non-reliable metric used by the authors to characterize the division of labour, partially by the absence of the precise definition of what behaviour counts as a "division of labour" in the considered system. I am also surprised by the authors' choice of the partitioning of their results between the main text and supplementary - there are some exciting findings buried in the supplementary. Below are the detailed comments to the authors.

Major comments

1. The metric of Duarte as in Eq.4 does not appear to me as a reliable measure of the division of labour. First, a complete specialization may result in a low value of the metric. Consider a completely specialized population, where individuals never switch tasks. However, tasks are distributed unequally: a fraction of the population $e \ll 1$ performs task one, and $1-e$ performs task two. Then $q = 1$, $p_1 = e$, $p_2 = 1-e$ and $D = 1/((1-e)^2 + e^2) - 1 \sim 2e \ll 1$. Second, an absence of specialization may result in a high value of the metric. If individuals switch between tasks every X steps, then $q = 1-1/X$, $p_1 = p_2 = 1/2$, and thus $D = 2(1-1/X) - 1 = 1 - 2/X$. At $X = 10$, individuals change the task every 10 steps - a much shorter period than the length of simulation ($T = 10'000$) or even the time needed for a well-fed individual to become hungry enough to go foraging ($(n_{\max} - \mu)/m_{\text{nur}} = 50$). Still, the metric there is as high as $D = 0.8$. I think that with a more reliable metric, the results of this study will be much more convincing.

2. The definition of the division of labour used in the paper ("the non-random association of group-living individuals with tasks to be performed") is not operational. The very definition of the model does not allow a stochastic task allocation: hungry individuals always go foraging, and well-fed individuals always do nursing. Even at the metric $D = 0$, the task allocation at the next time step can be accurately predicted from the state of the system at the end of the previous step - and hence it is not random by definition. So, using the definition above, every possible outcome of the model is a "division of labour". The manuscript needs at least a definition of the division of labour that does not contradict the presented results. A better option would be a discussion of the phenomenon: how a complete division of labour looks like, what its absence looks like, and what kinds of intermediary states exist.

3. The results of the nutrition-based sharing model presented in Fig.4b are actually the least surprising in the whole study. The very setup of the nutrition-based sharing leads to a "rich get richer" scenario for $s > 0$. Nothing else can evolve there. However, Fig.S3 in the appendix shows that some degree of the division of labour emerges at $s < 0$, where sharing promotes individuals to have close nutritional levels. And this is a counter-intuitive result and I am eager to know more about factors that drive individuals to keep some degree of inequality there.

Minor comment:

4. Model explains that individuals turn to forage when their resources drop below a threshold (II 255-258). However, the text does not explain how individuals turn back to nursing. The explanation in the "task choice" section is ambiguous. I assume that both tasks are assigned at each comparison of the nutritional level with a stochastic threshold: if an individual has low enough resources it goes foraging, otherwise, it stays and nurse.

Response to the comments of the reviewers:

We thank the two reviewers for providing detailed and constructive comments on our manuscript. We have revised the manuscript considerably in response to these comments. The main changes are:

- As requested by the reviewers, we revised the definition of ‘division of labour’.
- We provide a more detailed justification for the division-of-labour metric used in our study and address the concerns of Reviewer 2 on the usefulness and interpretation of this metric.
- As requested by Reviewer 1, we discuss our findings more broadly, linking them to the literature on task specialization in microorganisms and evolving robots. In particular, we discuss the role of phenotypic noise in our model in the light of that literature.

Reviewer 1:

This paper uses simulations to lay out different scenarios in which division of labour can emerge spontaneously from groups with identical individuals. They find that resource sharing is sufficient for task specialisation to occur between "foragers" and "nurses". They find that higher levels of task specialisation occur if: 1. there is a higher metabolic cost to foraging, 2. if there is a dominance hierarchy between individuals, and 3. if more nutritionally starved individuals receive more resources when foragers share.

This is a well developed model with some interesting findings. However, the framing of the model and the context of its structure and results within the wide body of theory could be improved. I make below several key suggestions for the framing of this paper.

First, I found the paper's definition of division of labour (line 35) to be unsatisfactory. The field is currently trying to move to more precise definitions (for instance their references 2 and 13) and the definition proposed here is both unintuitive and vague. For instance, no mention is made that individuals must perform distinct tasks. Connecting the results of this paper to other theory first requires a clear understanding of how the definition of division of labour in this paper may compare or contrast to the definition of other works. For instance, I think that they are using a more descriptive, less evolutionary definition of division of labour more similar to reference 2 than reference 13, but this distinction is not made clear in the manuscript. I would like this to be more clear/explicit.

We are pleased with the positive assessment of our model by the reviewer. We agree that our definition of ‘division of labour’ was unsatisfactory. In line with the reviewer’s suggestion, we now define division of labour as “within-individual consistency but between-individual variability in task choice among the members of a group” (first sentence of the introduction). We now justify (lines 221-229) why we do not use an “evolutionary definition” of division of labour: in our model, division of labour is not shaped by natural selection, but rather a byproduct of resource sharing. Yet, we also stress that our model can have evolutionary implications, as the spontaneously emerging division of labour can be the starting point of evolution by natural selection (lines 229-235).

Secondly, this paper focuses on previous theory that uses response-threshold models and does not discuss the wider body of theory using alternative mechanisms, of which response threshold models are just a subset. For instance, there has been recent work discussing the broader variety of mechanisms by which division of labour can evolve (Cooper et al. "The evolution of mechanisms to produce phenotypic heterogeneity in microorganisms." Nature Communications (2022)). In particular, there is also work on how division of labour can emerge from “phenotypic noise” (Ackermann et al. "Self-destructive cooperation mediated by phenotypic noise." Nature 454.7207 (2008): 987-990.).

Most notably, the paper "Evolution of self-organized task specialization in robot swarms." By Ferrante, et al. PLoS computational biology (2015) is not cited which seems to have many similarities in set-up and results to this work and so a thorough discussion of differences in assumptions and findings with this work is needed.

We thank the reviewer for directing our attention to these interesting papers. In a new section (lines 201-209), we now discuss microbial specialization and phenotypic noise. Indeed, our model also builds on phenotypic noise to break the symmetry between individuals. However, we also highlight that in our model phenotypic noise, on its own, does not lead to division of labour (as is shown in Fig. 2A). For this, resource sharing is required, a mechanism that we did not encounter in the microbial literature. We now also discuss the interesting study of Ferrante *et al.* (2015). We agree with the reviewer that the findings of that study are intriguing, but we disagree that there are "many similarities in set-up and results" to our work. The Ferrante model is an evolutionary model, while in our model division of labour emerges spontaneously, in the absence of evolution. Moreover, the agents in the Ferrante model do not share resources, and they do not have a 'state dynamics' (the systematic decrease in nutrition level with time) that is a characteristic feature of our model. In other words, the two models are very different and cannot easily be compared. However, one finding of the Ferrante study is relevant for our study as well: Ferrante *et al.* (2015) found that the evolution of division of labour is more efficient if a primordial form of division of labour (as the division of labour emerging in our model) already exists from the start. We now discuss this point in lines 230-235.

Third, I think that there are a couple key aspects of the model that are not discussed which would help to understand why the results emerge. First, there is an asymmetry between the roles built into the model where foragers share resources only with nurses and not other foragers. Do the results of the model work if foragers share benefits with all individuals regardless of their role? I think the connection to the above Ferrante paper would be clearer if this were expounded upon. Second, stochastic noise is built into the model by the internal threshold that individuals use to determine if they begin foraging. This seems to be a clear way that symmetry between individuals is broken in the model but is not sufficiently discussed. Does the model work without this stochastic component? In a sense this would make this mechanism a form of division of labour that is already expounded upon in the literature on phenotypic noise (the above Ackermann paper and Bistability in bacteria by Dubnau in Molecular Microbiology 2006). I think further discussion of this connection is warranted.

We would like to highlight that the asymmetry between the tasks in our model does not imply an asymmetry between individuals. We now made this more explicit by stating in the methods (lines 290-293): "Resources can be received through resource sharing by nursing individuals, which include previously foraging individuals that refilled their nutrition levels and switched back to nursing. Individuals thus have no pre-disposition to perform nursing or foraging tasks and all individuals can in principle share and receive resources." Thus, as the reviewer suggests, it already is the case that resources are shared with all individuals that are not currently foraging or processing resources.

As mentioned above, stochastic noise does indeed break the symmetry between individuals, but, in contrast to studies cited by the referee, noise does not lead to *consistent* differences between individuals. We now realize that our description of stochastic noise was ambiguous, and we have improved this description considerably (lines 282-285). Importantly, not the internal threshold is subject to noise in our model, but the individual's perception of the nutrition level. As noise is added each time the nutrition level is assessed (independently of earlier assessments), an individual will sometimes start foraging a bit earlier and sometimes a bit later than its true nutrition level indicates. In other words, noise may affect the timing of events throughout an individual's lifetime, but it does

not induce consistent inter-individual differences. Thanks to the reviewer's comments, we have now clarified our model description (lines 280-298). Moreover, we now discuss the implications of noise in our model (in comparison to the studies cited by the reviewer) in a separate paragraph (lines 201-209).

Finally, much of the theoretical literature in the area has asked why division of labour might evolve over uniform cooperation in social groups. This model suggests that division of labour should arise spontaneously from uniform cooperation due to resource/benefit sharing. Yet, we know empirically that this is not always the case. For instance, Pseudomonas aeruginosa, where any phenotypic heterogeneity in siderophore production arises from cheating not division of labour benefits (Jiricny et al. Fitness correlates with the extent of cheating in a bacterium. J. Evol. Biol. (2010)). Discussing why shared resources may not lead to division of labour in some cases is needed.

We now explain more clearly that our model does not include any evolution (also see our remarks above). Consequently, our model cannot address the question of why division of labour evolves over uniform cooperation. However, as we now also highlight more clearly (lines 230-235) our model proposes a mechanism from which the evolution of such division of labour, including cooperation and division of labour benefits, as mentioned by the reviewer, could be facilitated.

Reviewer 2:

Thank you for the opportunity to review the manuscript "Resource sharing leads to the emergence of division of labour" by J.J.Kreider and colleagues. In this work, the authors consider a population of agents switching tasks between nursing and foraging. Execution of both tasks consumes resources, which can be gained by performing foraging. Resources can be shared between individuals. Authors consider a number of well-designed scenarios of resource sharing: a negative control case of no sharing, the simplest equal sharing scenario, and two more elaborate cases - dominance-based and nutrition-based scenarios. Using this model, the authors perform simulations of the population dynamics and quantify the degree of the division of labour using the chosen metric. They show that the more elaborate scenarios of resource sharing results in more specialization. They also consider a number of variants of their model screening through the control parameters: group size and metabolic costs of the tasks.

The work addresses a fundamental question in biology - the emergence of specialization in a collective of initially identical organisms. The manuscript is well-written and very accessible. The model is simple enough to be easily comprehended but rich enough to address the problem in the focus of the study. Hence, there is a lot of merit in this work.

We are pleased with the generally positive assessment of our study by the reviewer.

However, do not find the presented results convincing: partially due to the non-reliable metric used by the authors to characterize the division of labour, partially by the absence of the precise definition of what behaviour counts as a "division of labour" in the considered system. I am also surprised by the authors' choice of the partitioning of their results between the main text and supplementary - there are some exciting findings buried in the supplementary. Below are the detailed comments to the authors.

We will address these more critical comments in detail below. For convenience of explanation, we have swapped the first two major comments of the reviewer.

Major comments

2. The definition of the division of labour used in the paper ("the non-random association of group-living individuals with tasks to be performed") is not operational. The very definition of the model does not allow a stochastic task allocation: hungry individuals always go foraging, and well-fed individuals always do nursing. Even at the metric $D = 0$, the task allocation at the next time step can be accurately predicted from the state of the system at the end of the previous step - and hence it is not random by definition. So, using the definition above, every possible outcome of the model is a "division of labour". The manuscript needs at least a definition of the division of labour that does not contradict the presented results. A better option would be a discussion of the phenomenon: how a complete division of labour looks like, what its absence looks like, and what kinds of intermediary states exist.

We agree with the reviewer that our definition of 'division of labour' was unsatisfactory. As indicated in our response to Reviewer 1, we now use one of the standard definitions of division of labour as "within-individual consistency but between-individual variability in task choice among the members of a group" (lines 35-36). This definition is more operational than the earlier one, but, as highlighted in the Methods section (lines 331-337), it is based on two properties (within-individual consistency and between-individual variability) that cannot easily be quantified by a single metric. This has been the subject of quite some debate in the literature (some of the articles on this matter are cited in our manuscript), resulting in different metrics – some focusing on within-individual consistency and others on between-individual variability (see Gorelick *et al.* 2004, *AmNat*, for a detailed discussion). In our study, we quantified division of labour by the four metrics that are most often encountered in the literature. Supplementary Fig. S1 shows that, in our study, these four metrics yield similar results in all scenarios considered, which indicates that our results are not just artefacts of the division-of-labour metric used. In the main text, we continue to use the metric D introduced by Duarte *et al.* (2012), as it is easy to calculate and easy to explain to a non-expert audience. We have improved the explanation of D and also pointed out some limitations of this metric (lines 345-351).

1. The metric of Duarte as in Eq.4 does not appear to me as a reliable measure of the division of labour. First, a complete specialization may result in a low value of the metric. Consider a completely specialized population, where individuals never switch tasks. However, tasks are distributed unequally: a fraction of the population $e \ll 1$ performs task one, and $1-e$ performs task two. Then $q = 1$, $p_1 = e$, $p_2 = 1-e$ and $D = 1/((1-e)^2 + e^2) - 1 \sim 2e \ll 1$. Second, an absence of specialization may result in a high value of the metric. If individuals switch between tasks every X steps, then $q = 1-1/X$, $p_1 = p_2 = 1/2$, and thus $D = 2(1-1/X) - 1 = 1 - 2/X$. At $X = 10$, individuals change the task every 10 steps - a much shorter period than the length of simulation ($T = 10'000$) or even the time needed for a well-fed individual to become hungry enough to go foraging $(n_{max} - \mu)/m_{nur} = 50$. Still, the metric there is as high as $D = 0.8$. I think that with a more reliable metric, the results of this study will be much more convincing.

We appreciate how thoroughly the reviewer engaged with the division-of-labour metric D used in our study. Regarding the first point, the reviewer is entirely correct. D only ranges from -1 to +1 if $p_1 = p_2$; otherwise it ranges from -1 to $2p_1p_2/(p_1^2 + p_2^2)$. In other words, D is a convenient measure, but it has the drawback that it is mainly useful in situations where $p_1 \approx p_2$. Fortunately, the latter is the case in all scenarios considered in our study. We now explain the scope and limitations of D in more detail in the Methods section (lines 345-351) and refer to this explanation in the Results section (line 92). In Fig. S1, we furthermore show that the metric D leads to the same conclusions as three other commonly used measures of division of labour.

Regarding the second point, we are sorry for not being sufficiently explicit when explaining how the division-of-labour metric was calculated. The reviewer's concerns would be entirely correct when the metric D were calculated on a per-time-step basis. In that case, our assumption that a foraging trip takes five time steps would imply that – by assumption – a foraging individual would perform the same task at least five times in a row. The reviewer is completely right that such a procedure would inflate the metric D , making it almost meaningless. However, this is not how we actually quantified D . D is not quantified on a per-time-step basis, but on a per-decision-situation basis. Taking foraging, we assess *at the end of a foraging trip* whether the individual will start a new foraging trip or switch to nursing. Taking the reviewer's example, a forager switching to nursing every 10 decision situations would be out foraging for 50 (=10x5) time steps. To us, it makes perfect sense that a situation like this is indicated by a D -value of 0.8. We are grateful that the reviewer pointed out this potential misinterpretation of D . In the revised version, the calculation of D is now explained in more detail (lines 338-351).

3. The results of the nutrition-based sharing model presented in Fig.4b are actually the least surprising in the whole study. The very setup of the nutrition-based sharing leads to a "rich get richer" scenario for $s>0$. Nothing else can evolve there. However, Fig.S3 in the appendix shows that some degree of the division of labour emerges at $s<0$, where sharing promotes individuals to have close nutritional levels. And this is a counter-intuitive result and I am eager to know more about factors that drive individuals to keep some degree of inequality there.

We are pleased that the reviewer considers this result interesting, and we added an explanation to the Results section (lines 151-156). We also highlight that this result is – in a sense – already foreshadowed by Fig. 3 in the main text, where (a low level of) division of labour even emerges if nursing is associated with higher metabolic costs than foraging, thus where nursing individuals are predisposed towards malnourishment compared to foraging individuals. For this reason (and in view of the space limitations imposed by the journal) we decided to keep Fig. S3 in the Supplement.

Minor comment:

4. Model explains that individuals turn to forage when their resources drop below a threshold (ll 255-258). However, the text does not explain how individuals turn back to nursing. The explanation in the "task choice" section is ambiguous. I assume that both tasks are assigned at each comparison of the nutritional level with a stochastic threshold: if an individual has low enough resources it goes foraging, otherwise, it stays and nurse.

We revised the corresponding part of the Methods section (lines 285-298) and now explain in more detail how individuals choose tasks.

Reviewers' Comments:

Reviewer #1:

Remarks to the Author:

This is a well written manuscript with a well designed model. However, I have two key outstanding issues that remain.

First, the authors state that the feedback mechanism of resource sharing has not been previously explored in the theoretical literature on division of labour (lines 207-209). However, this is not the case. For instance, the "shareability" of a good is at the core of the results in Schiessl, Konstanze T., et al. "Individual-versus group-optimality in the production of secreted bacterial compounds." *Evolution* 73.4 (2019): 675-688. Similarly for the trait "sociality" in Cooper, Guy A., and Stuart A. West. "Division of labour and the evolution of extreme specialization." *Nature ecology & evolution* 2.7 (2018): 1161-1167. I think that it is well established in the literature that division of labour cannot arise in asocial systems (with no sharing) and that higher levels of trait sociality/good shareability favours division of labour. What is different and interesting about this model in comparison to previous treatments lies in the details of how the resource is shared asymmetrically. Yet, I do think that a very similar task asymmetry is present in the Ferrante paper I cited in my last review.

Second, I think that the similarities between this model and previous response-threshold models is not made clear enough (lines 47-49) and (lines 60-63). In most response threshold models, there will be stochastic inter-individual variability in a response threshold which can break the symmetry and lead to division of labour. Importantly, this inter-individual difference can arise from phenotypic noise rather than any deeper differences between individuals. This model is different from response threshold models, in that individuals do not have different stochastic thresholds. Instead, individuals perceive the signal stochastically when comparing it to their common threshold, which ultimately has the exact same effect in breaking symmetry. I think that this setup is actually very similar to response threshold models.

The authors present resource sharing and the pre-existing inter-individual differences of response threshold models as the key motivation for this model. However, I am not convinced that their model is really that different from previous treatments in this regard, which unfortunately undermines the novelty of this work.

Reviewer #2:

Remarks to the Author:

In the updated manuscript, the authors resolved concerns raised in the previous version. The definition of phenomena and descriptions of their methods are much improved.

Still, there are a few issues that arise after reading this manuscript.

1. Authors describe that the division of labour metric D is measured on the decision event basis. For a continuously foraging individual, decisions are made every $t_{\text{for}} = 5$ time steps. But for a continuously nursing individual, decisions cannot be made for $t_{\text{nur}} = 5$ time steps after feeding. What is not yet explained is how frequent is the decision-making of a nursing individual after t_{nur} has passed and it has enough resources to continue nursing. Also, it is not clear, how often would a forager wait until the next decision event, if it has more than 50 food after sharing: on the one hand it turns into a nursing role and rules for waiting of a nursing individual should apply, on the other hand, it didn't receive any food by means of sharing, so the waiting time t_{nur} is not applicable. I would like the authors to clarify the rules of decision-making frequency.

2. In the absence of explanation, I assume that after digesting food for a period t_{nur} , a nursing

individual makes a decision every simulation step until the resource pool falls low or another feeding event happens. Maybe this is not how the model works, but if so, nursing individuals can make decisions at a much higher rate than foraging individuals. This might have a strong effect on the model outcome. With the parameters used in the simulation, a well-supplied nursing individual contributes as many decision events to the pool used to compute D , as 5 foraging individuals (who can only make a decision every $t_{\text{for}} = 5$ steps). This way, the metric D can be heavily biased towards the decisions made by nursing individuals alone, while decisions of foragers have little weight. First, I am curious if such an effect is in action. If yes, then the metric D does not represent the state of the whole population as the authors claim. So, if the decision pool represents the decisions of only a sub-population, what is a correct interpretation of observed results (specialization of the nursing sub-population)? For the first question, it might help to have a figure comparing the fraction of the number of nursing individuals with the fraction of decision events produced by them.

Response to the comments of the reviewers:

We thank the reviewers for their comments on our manuscript. We have revised the manuscript following their suggestions and think that this has improved the manuscript. Please find our detailed response to the reviewers' comments below.

Reviewer 1:

*This is a well written manuscript with a well designed model. However, I have two key outstanding issues that remain. First, the authors state that the feedback mechanism of resource sharing has not been previously explored in the theoretical literature on division of labour (lines 207-209). However, this is not the case. For instance, the "shareability" of a good is at the core of the results in Schiessl, Konstanze T., et al. "Individual-versus group-optimality in the production of secreted bacterial compounds." *Evolution* 73.4 (2019): 675-688. Similarly for the trait "sociality" in Cooper, Guy A., and Stuart A. West. "Division of labour and the evolution of extreme specialization." *Nature ecology & evolution* 2.7 (2018): 1161-1167. I think that it is well established in the literature that division of labour cannot arise in asocial systems (with no sharing) and that higher levels of trait sociality/good shareability favours division of labour. What is different and interesting about this model in comparison to previous treatments lies in the details of how the resource is shared asymmetrically. Yet, I do think that a very similar task asymmetry is present in the Ferrante paper I cited in my last review.*

Our statement on feedback mechanisms (previously lines 207-209) was indeed misleading, and we are grateful to the reviewer for pointing this out. Of course, the shared use of globally available resources also plays a key role in other division-of-labour models. However, our model makes the crucial difference that resource sharing acts at the individual level and affects an individual-level property (nutrition level). We have now clarified this (lines 219-223). Furthermore, we also incorporated the argument of the reviewer that the shareability of resources/benefits is essential for the evolution of division of labour (lines 240-242). However, we also stress that our model is not an evolutionary model but a mechanistic model for the self-organized emergence of division of labour. Although resource/benefit sharing is essential for the evolution of division of labour in the evolutionary models that the reviewer mentions, these models do not explain how division of labour originates mechanistically. In contrast, we propose a mechanism that leads to the spontaneous emergence of division of labour, irrespective of whether resource sharing provides a fitness advantage to the social group or not. This is now highlighted more clearly in the manuscript (lines 242-250).

Second, I think that the similarities between this model and previous response-threshold models is not made clear enough (lines 47-49) and (lines 60-63). In most response threshold models, there will be stochastic inter-individual variability in a response threshold which can break the symmetry and lead to division of labour. Importantly, this inter-individual difference can arise from phenotypic noise rather than any deeper differences between individuals. This model is different from response threshold models, in that individuals do not have different stochastic thresholds. Instead, individuals perceive the signal stochastically when comparing it to their common threshold, which ultimately has the exact same effect in breaking symmetry. I think that this setup is actually very similar to response threshold models.

The authors present resource sharing and the pre-existing inter-individual differences of response threshold models as the key motivation for this model. However, I am not convinced that their model is really that different from previous treatments in this regard, which unfortunately undermines the novelty of this work.

We have added a new section to the discussion where we explain the differences between response threshold models and our model in more detail (lines 199-212). We incorporated the suggestion by

the reviewer that interindividual differences between response thresholds could emerge from phenotypic noise (lines 200-202). However, we do not agree with the reviewer that stochasticity in response thresholds is ultimately the same as stochasticity in the cue perceived by the individuals. We now illustrate how these two alternatives – reflected by response threshold models and our model – are different and suggest an experimental setup that could be used to elucidate which of the mechanisms is in place to lead to the emergence of division of labour (lines 204-212). If a mechanism as suggested by response threshold models is in place, individuals would be likely to perform the same task again when moved to another group. However, if a mechanism as suggested by our model is in place, then this should not be the case because individuals are initially identical and the task, they specialize in, is emergent from the model dynamics.

Reviewer 2:

In the updated manuscript, the authors resolved concerns raised in the previous version. The definition of phenomena and descriptions of their methods are much improved. Still, there are a few issues that arise after reading this manuscript.

1. Authors describe that the division of labour metric D is measured on the decision event basis. For a continuously foraging individual, decisions are made every $t_{\text{for}} = 5$ time steps. But for a continuously nursing individual, decisions cannot be made for $t_{\text{nur}} = 5$ time steps after feeding. What is not yet explained is how frequent is the decision-making of a nursing individual after t_{nur} has passed and it has enough resources to continue nursing. Also, it is not clear, how often would a forager wait until the next decision event, if it has more than 50 food after sharing: on the one hand it turns into a nursing role and rules for waiting of a nursing individual should apply, on the other hand, it didn't receive any food by means of sharing, so the waiting time t_{nur} is not applicable. I would like the authors to clarify the rules of decision-making frequency.

We have added a new figure to the supplementary materials that explains in detail when task choice situations occur (Fig. S5; as also requested by the reviewer in the comment below). In the figure caption, we explain: “As our model includes two tasks, four transitions between tasks are possible. (1) *Foraging to foraging*: Foraging individuals share resources with nursing individuals and subsequently forage again if their nutrition level is so low that they perceive a hunger signal. (2) *Foraging to nursing*: If the foraging individual does not perceive a hunger signal after food sharing, it switches to nursing. (3) *Nursing to nursing*: Nursing individuals that obtain resources from foraging individuals process these resources and subsequently nurse again if their nutrition level is sufficiently high so that they do not perceive a hunger signal. (4) *Nursing to foraging*: Nursing individuals switch to foraging if they perceive a hunger signal. This can happen after nursing individuals obtained and processed food or before they obtain food from a foraging individual.”

2. In the absence of explanation, I assume that after digesting food for a period t_{nur} , a nursing individual makes a decision every simulation step until the resource pool falls low or another feeding event happens. Maybe this is not how the model works, but if so, nursing individuals can make decisions at a much higher rate than foraging individuals. This might have a strong effect on the model outcome. With the parameters used in the simulation, a well-supplied nursing individual contributes as many decision events to the pool used to compute D , as 5 foraging individuals (who can only make a decision every $t_{\text{for}} = 5$ steps). This way, the metric D can be heavily biased towards the decisions made by nursing individuals alone, while decisions of foragers have little weight. First, I am curious if such an effect is in action. If yes, then the metric D does not represent the state of the whole population as the authors claim. So, if the decision pool represents the decisions of only a sub-population, what is a correct interpretation of observed results (specialization of the nursing sub-population)? For the first question, it might help to have a figure comparing the fraction of the number of nursing individuals with the fraction of decision events produced by them.

We are sorry that our explanation has been ambiguous before and we hope that Fig. S5 clarifies when task choice situations, which are at the basis for the calculation of the division-of-labour metric, occur. The reviewer was concerned about an asymmetric contribution of task choice situations between the castes to the division-of-labour metric. We have added to the figure caption of Fig. S5 that this was not the case: "In all of the main model scenarios, nursing and foraging individuals contributed an approximately equal number of task choice situations to the calculation of the division-of-labour metric." In fact, the percentage of such situations was highly similar (between 48% and 52% for both categories) in the main scenarios discussed in the manuscript.